# Co-Mixup: Saliency Guided Joint Mixup with Supermodular Diversity

**Jang-Hyun Kim, Wonho Choo, Hosan Jeong, Hyun Oh Song**
Department of Computer Science and Engineering, Seoul National University
Neural Processing Research Center
`{janghyun,wonho.choo,grazinglion,hyunoh}@mllab.snu.ac.kr`

## Abstract

While deep neural networks show great performance on fitting to the training distribution, improving the networks' generalization performance to the test distribution and robustness to the sensitivity to input perturbations still remain as a challenge. Although a number of mixup based augmentation strategies have been proposed to partially address them, it remains unclear as to how to best utilize the supervisory signal within each input data for mixup from the optimization perspective. We propose a new perspective on batch mixup and formulate the optimal construction of a batch of mixup data maximizing the data saliency measure of each individual mixup data and encouraging the supermodular diversity among the constructed mixup data. This leads to a novel discrete optimization problem minimizing the difference between submodular functions. We also propose an efficient modular approximation based iterative submodular minimization algorithm for efficient mixup computation per each minibatch suitable for minibatch based neural network training. Our experiments show the proposed method achieves the state of the art generalization, calibration, and weakly supervised localization results compared to other mixup methods. The source code is available at `https://github.com/snu-mllab/Co-Mixup`.

## 1 Introduction

Deep neural networks have been applied to a wide range of artificial intelligence tasks such as computer vision, natural language processing, and signal processing with remarkable performance (Ren et al., 2015; Devlin et al., 2018; Oord et al., 2016). However, it has been shown that neural networks have excessive representation capability and can even fit random data (Zhang et al., 2016). Due to these characteristics, the neural networks can easily overfit to training data and show a large generalization gap when tested on previously unseen data.

To improve the generalization performance of the neural networks, a body of research has been proposed to develop regularizers based on priors or to augment the training data with task-dependent transforms (Bishop, 2006; Cubuk et al., 2019). Recently, a new task-independent data augmentation technique, called *mixup*, has been proposed (Zhang et al., 2018). The original mixup, called *Input Mixup*, linearly interpolates a given pair of input data and can be easily applied to various data and tasks, improving the generalization performance and robustness of neural networks. Other mixup methods, such as *manifold mixup* (Verma et al., 2019) or *CutMix* (Yun et al., 2019), have also been proposed addressing different ways to mix a given pair of input data. *Puzzle Mix* (Kim et al., 2020) utilizes saliency information and local statistics to ensure mixup data to have rich supervisory signals.

However, these approaches only consider mixing a given random pair of input data and do not fully utilize the rich informative supervisory signal in training data including collection of object saliency, relative arrangement, etc. In this work, we simultaneously consider mix-matching different salient regions among all input data so that each generated mixup example accumulates as many salient regions from multiple input data as possible while ensuring

---

Correspondence to: Hyun Oh Song.

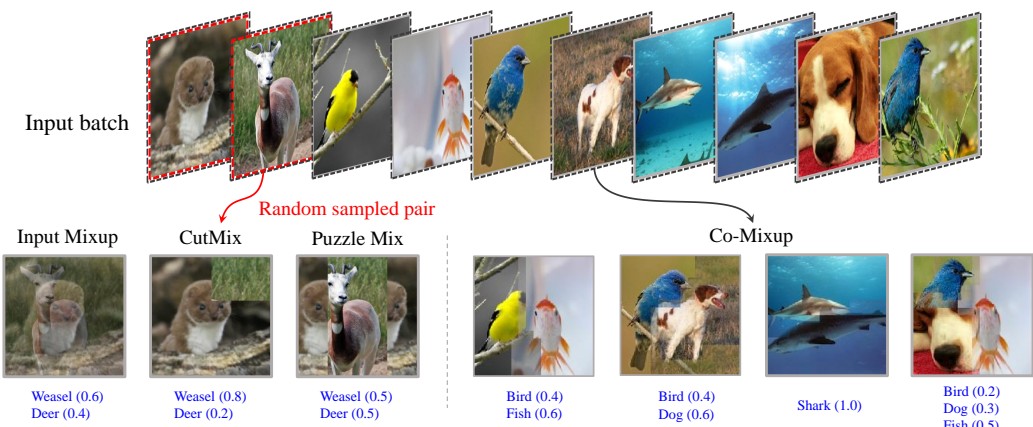

Figure 1: Example comparison of existing mixup methods and the proposed Co-Mixup. We provide more samples in Appendix H.

diversity among the generated mixup examples. To this end, we propose a novel optimization problem that maximizes the saliency measure of each individual mixup example while encouraging diversity among them collectively. This formulation results in a novel discrete submodular-supermodular objective. We also propose a practical modular approximation method for the supermodular term and present an efficient iterative submodular minimization algorithm suitable for minibatch-based mixup for neural network training. As illustrated in the Figure 1, while the proposed method, *Co-Mixup*, mix-matches the collection of salient regions utilizing inter-arrangements among input data, the existing methods do not consider the saliency information (Input Mixup & CutMix) or disassemble salient parts (Puzzle Mix).

We verify the performance of the proposed method by training classifiers on CIFAR-100, Tiny-ImageNet, ImageNet, and the Google commands dataset (Krizhevsky et al., 2009; Chrabaszcz et al., 2017; Deng et al., 2009; Warden, 2017). Our experiments show the models trained with Co-Mixup achieve the state of the performance compared to other mixup baselines. In addition to the generalization experiment, we conduct weakly-supervised object localization and robustness tasks and confirm Co-Mixup outperforms other mixup baselines.

## 2 RELATED WORKS

**Mixup**  Data augmentation has been widely used to prevent deep neural networks from over-fitting to the training data (Bishop, 1995). The majority of conventional augmentation methods generate new data by applying transformations depending on the data type or the target task (Cubuk et al., 2019). Zhang et al. (2018) proposed *mixup*, which can be independently applied to various data types and tasks, and improves generalization and robustness of deep neural networks. *Input mixup* (Zhang et al., 2018) linearly interpolates between two input data and utilizes the mixed data with the corresponding soft label for training. Following this work, *manifold mixup* (Verma et al., 2019) applies the mixup in the hidden feature space, and *CutMix* (Yun et al., 2019) suggests a spatial copy and paste based mixup strategy on images. Guo et al. (2019) trains an additional neural network to optimize a mixing ratio. *Puzzle Mix* (Kim et al., 2020) proposes a mixup method based on saliency and local statistics of the given data. In this paper, we propose a discrete optimization-based mixup method simultaneously finding the best combination of collections of salient regions among all input data while encouraging diversity among the generated mixup examples.

**Saliency**  The seminal work from Simonyan et al. (2013) generates a saliency map using a pre-trained neural network classifier without any additional training of the network. Following the work, measuring the saliency of data using neural networks has been studied to obtain a more precise saliency map (Zhao et al., 2015; Wang et al., 2015) or to reduce the saliency computation cost (Zhou et al., 2016; Selvaraju et al., 2017). The saliency information is widely applied to the tasks in various domains, such as object segmentation or speech recognition (Jung and Kim, 2011; Kalinli and Narayanan, 2007).

**Submodular-Supermodular optimization** A submodular (supermodular) function is a set function with diminishing (increasing) returns property (Narasimhan and Bilmes, 2005). It is known that any set function can be expressed as the sum of a submodular and supermodular function (Lovász, 1983), called BP function. Various problems in machine learning can be naturally formulated as BP functions (Fujishige, 2005), but it is known to be NP-hard (Lovász, 1983). Therefore, approximate algorithms based on modular approximations of submodular or supermodular terms have been developed (Iyer and Bilmes, 2012). Our formulation falls into a category of BP function consisting of smoothness function within a mixed output (submodular) and a diversity function among the mixup outputs (supermodular).

## 3 Preliminary

Existing mixup methods return $\{h(x_1, x_{i(1)}), \ldots, h(x_m, x_{i(m)})\}$ for given input data $\{x_1, \ldots, x_m\}$, where $h : \mathcal{X} \times \mathcal{X} \to \mathcal{X}$ is a mixup function and $(i(1), \ldots, i(m))$ is a random permutation of the data indices. In the case of input mixup, $h(x, x')$ is $\lambda x + (1 - \lambda)x'$, where $\lambda \in [0, 1]$ is a random mixing ratio. Manifold mixup applies input mixup in the hidden feature space, and CutMix uses $h(x, x') = \mathbb{1}_B \odot x + (1 - \mathbb{1}_B) \odot x'$, where $\mathbb{1}_B$ is a binary rectangular-shape mask for an image $x$ and $\odot$ represents the element-wise product. Puzzle Mix defines $h(x, x')$ as $z \odot \Pi^\intercal x + (1 - z) \odot \Pi'^\intercal x'$, where $\Pi$ is a transport plan and $z$ is a discrete mask. In detail, for $x \in \mathbb{R}^n$, $\Pi \in \{0, 1\}^n$ and $z \in \mathcal{L}^n$ for $\mathcal{L} = \{\frac{l}{L} \mid l = 0, 1, \ldots, L\}$.

In this work, we extend the existing mixup functions as $h : \mathcal{X}^m \to \mathcal{X}^{m'}$ which performs mixup on a collection of input data and returns another collection. Let $x_B \in \mathbb{R}^{m \times n}$ denote the batch of input data in matrix form. Then, our proposed mixup function is

$$h(x_B) = \big(g(z_1 \odot x_B), \ldots, g(z_{m'} \odot x_B)\big),$$

where $z_j \in \mathcal{L}^{m \times n}$ for $j = 1, \ldots, m'$ with $\mathcal{L} = \{\frac{l}{L} \mid l = 0, 1, \ldots, L\}$ and $g : \mathbb{R}^{m \times n} \to \mathbb{R}^n$ returns a column-wise sum of a given matrix. Note that, the $k^{\text{th}}$ column of $z_j$, denoted as $z_{j,k} \in \mathcal{L}^m$, can be interpreted as the mixing ratio among $m$ inputs at the $k^{\text{th}}$ location. Also, we enforce $\|z_{j,k}\|_1 = 1$ to maintain the overall statistics of the given input batch. Given the one-hot target labels $y_B \in \{0, 1\}^{m \times C}$ of the input data with $C$ classes, we generate soft target labels for mixup data as $y_B^\intercal \tilde{o}_j$ for $j = 1, \ldots, m'$, where $\tilde{o}_j = \frac{1}{n} \sum_{k=1}^n z_{j,k} \in [0, 1]^m$ represents the input source ratio of the $j^{\text{th}}$ mixup data. We train models to estimate the soft target labels by minimizing the cross-entropy loss.

## 4 Method

### 4.1 Objective

**Saliency** Our main objective is to maximize the saliency measure of mixup data while maintaining the local smoothness of data, *i.e.*, spatially nearby patches in a natural image look similar, temporally adjacent signals have similar spectrum in speech, etc. (Kim et al., 2020). As we can see from CutMix in Figure 1, disregarding saliency can give a misleading supervisory signal by generating mixup data that does not match with the target soft label. While the existing mixup methods only consider the mixup between two inputs, we generalize the number of inputs $m$ to any positive integer. Note, each $k^{\text{th}}$ location of outputs has $m$ candidate sources from the inputs. We model the unary labeling cost as the negative value of the saliency, and denote the cost vector at the $k^{\text{th}}$ location as $c_k \in \mathbb{R}^m$. For the saliency measure, we calculate the gradient values of training loss with respect to the input and measure $\ell_2$ norm of the gradient values across input channels (Simonyan et al., 2013; Kim et al., 2020). Note that this method does not require any additional architecture dependent modules for saliency calculation. In addition to the unary cost, we encourage adjacent locations to have similar labels for the smoothness of each mixup data. In summary, the objective can be formulated as follows:

$$\sum_{j=1}^{m'} \sum_{k=1}^n c_k^\intercal z_{j,k} + \beta \sum_{j=1}^{m'} \sum_{(k,k') \in \mathcal{N}} (1 - z_{j,k}^\intercal z_{j,k'}) - \eta \sum_{j=1}^{m'} \sum_{k=1}^n \log p(z_{j,k}),$$

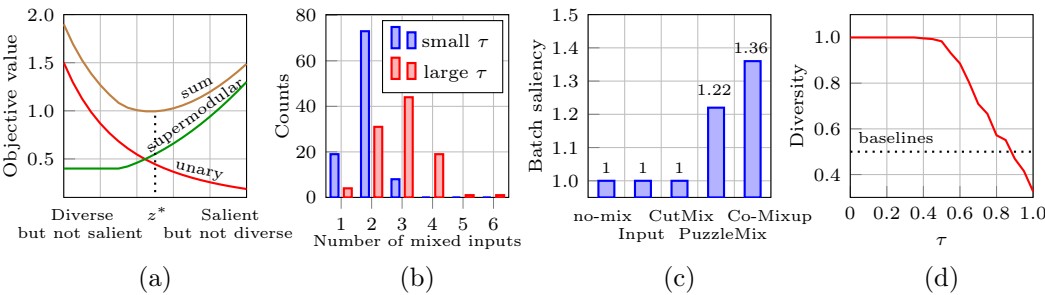

Figure 2: (a) Analysis of our BP optimization problem. The x-axis is a one-dimensional arrangement of solutions: The mixed output is more salient but not diverse towards the right and less salient but diverse on the left. The unary term (red) decreases towards the right side of the axis, while the supermodular term (green) increases. By optimizing the sum of the two terms (brown), we obtain the balanced output $z^*$. (b) A histogram of the number of inputs mixed for each output given a batch of 100 examples from the ImageNet dataset. As $\tau$ increases, more inputs are used to create each output on average. (c) Mean batch saliency measurement of a batch of mixup data using the ImageNet dataset. We normalize the saliency measure of each input to sum up to 1. (d) Diversity measurement of a batch of mixup data. We calculate the diversity as $1 - \sum_j \sum_{j' \neq j} \tilde{o}_j^\mathsf{T} \tilde{o}_{j'}/m$, where $\tilde{o}_j = o_j/\|o_j\|_1$. We can control the diversity among Co-Mixup data (red) and find the optimum by controlling $\tau$.

where the prior $p$ is given by $z_{j,k} \sim \frac{1}{L}Multi(L, \lambda)$ with $\lambda = (\lambda_1, \ldots, \lambda_m) \sim Dirichlet(\alpha, \ldots, \alpha)$, which is a generalization of the mixing ratio distribution of Zhang et al. (2018), and $\mathcal{N}$ denotes a set of adjacent locations (*i.e.*, neighboring image patches in vision, subsequent spectrums in speech, etc.).

**Diversity**  Note that the naive generalization above leads to the identical outputs because the objective is separable and identical for each output. In order to obtain diverse mixup outputs, we model a similarity penalty between outputs. First, we represent the input source information of the $j^{\text{th}}$ output by aggregating assigned labels as $\sum_{k=1}^n z_{j,k}$. For simplicity, let us denote $\sum_{k=1}^n z_{j,k}$ as $o_j$. Then, we measure the similarity between $o_j$'s by using the inner-product on $\mathbb{R}^m$. In addition to the input source similarity between outputs, we model the compatibility between input sources, represented as a symmetric matrix $A_c \in \mathbb{R}_+^{m \times m}$. Specifically, $A_c[i_1, i_2]$ quantifies the degree to which input $i_1$ and $i_2$ are suitable to be mixed together. In summary, we use inner-product on $A = (1 - \omega)I + \omega A_c$ for $\omega \in [0, 1]$, resulting in a supermodular penalty term. Note that, by minimizing $\langle o_j, o_{j'} \rangle_A = o_j^\mathsf{T} A o_{j'}$, $\forall j \neq j'$, we penalize output mixup examples with similar input sources and encourage each individual mixup examples to have high compatibility within. In this work, we measure the distance between locations of salient objects in each input and use the distance matrix $A_c[i, j] = \|\text{argmax}_k s_i[k] - \text{argmax}_k s_j[k]\|_1$, where $s_i$ is the saliency map of the $i^{\text{th}}$ input and $k$ is a location index (*e.g.*, $k$ is a 2-D index for image data). From now on, we denote this inner-product term as the *compatibility* term.

**Over-penalization**  The conventional mixup methods perform mixup as many as the number of examples in a given mini-batch. In our setting, this is the case when $m = m'$. However, the compatibility penalty between outputs is influenced by the pigeonhole principle. For example, suppose the first output consists of two inputs. Then, the inputs must be used again for the remaining $m' - 1$ outputs, or only $m - 2$ inputs can be used. In the latter case, the number of available inputs ($m - 2$) is less than the outputs ($m' - 1$), and thus, the same input must be used more than twice. Empirically, we found that the remaining compatibility term above over-penalizes the optimization so that a substantial portion of outputs are returned as singletons without any mixup. To mitigate the over-penalization issue, we apply clipping to the compatibility penalty term. Specifically, we model the objective so that no extra penalty would occur when the compatibility among outputs is below a certain level.

Now we present our main objective as following:

$$z^* = \operatorname*{argmin}_{z_{j,k} \in \mathcal{L}^m, \; \|z_{j,k}\|_1 = 1} f(z),$$

where

$$f(z) := \sum_{j=1}^{m'} \sum_{k=1}^{n} c_k^\mathsf{T} z_{j,k} + \beta \sum_{j=1}^{m'} \sum_{(k,k') \in \mathcal{N}} (1 - z_{j,k}^\mathsf{T} z_{j,k'}) \tag{1}$$

$$+ \gamma \underbrace{\max \left\{ \tau, \; \sum_{j=1}^{m'} \sum_{j' \neq j}^{m'} \left( \sum_{k=1}^{n} z_{j,k} \right)^\mathsf{T} A \left( \sum_{k=1}^{n} z_{j',k} \right) \right\}}_{= f_c(z)} - \eta \sum_{j=1}^{m'} \sum_{k=1}^{n} \log p(z_{j,k}).$$

In Figure 2, we describe the properties of the BP optimization problem of Equation (1) and statistics of the resulting mixup data. Next, we verify the supermodularity of the compatibility term. We first extend the definition of the submodularity of a multi-label function as follows (Windheuser et al., 2012).

**Definition 1.** *For a given label set $\mathcal{L}$, a function $s : \mathcal{L}^m \times \mathcal{L}^m \to \mathbb{R}$ is pairwise submodular, if $\forall x, x' \in \mathcal{L}^m$, $s(x, x) + s(x', x') \leq s(x, x') + s(x', x)$. A function $s$ is pairwise supermodular, if $-s$ is pairwise submodular.*

**Proposition 1.** *The compatibility term $f_c$ in Equation (1) is pairwise supermodular for every pair of $(z_{j_1,k}, z_{j_2,k})$ if $A$ is positive semi-definite.*

*Proof.* See Appendix B.1. □

Finally note that, $A = (1 - \omega)I + \omega A_c$, where $A_c$ is a symmetric matrix. By using spectral decomposition, $A_c$ can be represented as $UDU^\mathsf{T}$, where $D$ is a diagonal matrix and $U^\mathsf{T} U = UU^\mathsf{T} = I$. Then, $A = U((1 - \omega)I + \omega D)U^\mathsf{T}$, and thus for small $\omega > 0$, we can guarantee $A$ to be positive semi-definite.

## 4.2 ALGORITHM

Our main objective consists of modular (*unary, prior*), submodular (*smoothness*), and supermodular (*compatibility*) terms. To optimize the main objective, we employ the submodular-supermodular procedure by iteratively approximating the supermodular term as a modular function (Narasimhan and Bilmes, 2005). Note that $z_j$ represents the labeling of the $j^{\text{th}}$ output and $o_j$ represents the aggregated input source information of the $j^{\text{th}}$ output, $\sum_{k=1}^{n} z_{j,k}$. Before introducing our algorithm, we first inspect the simpler case without clipping.

**Proposition 2.** *The compatibility term $f_c$ without clipping is modular with respect to $z_j$.*

*Proof.* Note, $A$ is a positive symmetric matrix by the definition. Then, for an index $j_0$, we can represent $f_c$ without clipping in terms of $o_{j_0}$ as $\sum_{j=1}^{m'} \sum_{j'=1, j' \neq j}^{m'} o_j^\mathsf{T} A o_{j'} = 2 \sum_{j=1, j \neq j_0}^{m'} o_j^\mathsf{T} A o_{j_0} + \sum_{j=1, j \neq j_0}^{m'} \sum_{j'=1, j' \notin \{j_0, j\}}^{m'} o_j^\mathsf{T} A o_{j'} = (2 \sum_{j=1, j \neq j_0}^{m'} A o_j)^\mathsf{T} o_{j_0} + c = v_{-j_0}^\mathsf{T} o_{j_0} + c$, where $v_{-j_0} \in \mathbb{R}^m$ and $c \in \mathbb{R}$ are values independent with $o_{j_0}$. Finally, $v_{-j_0}^\mathsf{T} o_{j_0} + c = \sum_{k=1}^{n} v_{-j_0}^\mathsf{T} z_{j_0,k} + c$ is a modular function of $z_{j_0}$. □

By Proposition 2, we can apply a submodular minimization algorithm to optimize the objective with respect to $z_j$ when there is no clipping. Thus, we can optimize the main objective without clipping in coordinate descent fashion (Wright, 2015). For the case with clipping, we modularize the supermodular compatibility term under the following criteria:

1. The modularized function value should increase as the compatibility across outputs increases.
2. The modularized function should not apply an extra penalty for the compatibility below a certain level.

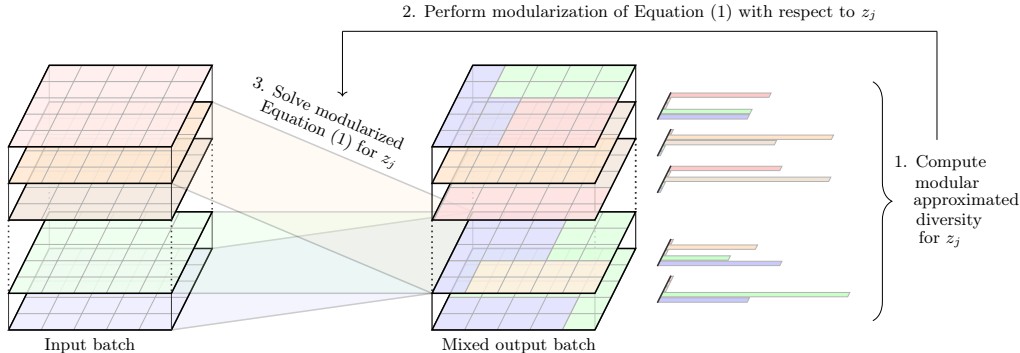

Figure 3: Visualization of the proposed mixup procedure. For a given batch of input data (left), a batch of mixup data (right) is generated, which mix-matches different salient regions among the input data while preserving the diversity among the mixup examples. The histograms on the right represent the input source information of each mixup data ($o_j$).

Borrowing the notation from the proof in Proposition 2, for an index $j$, $f_c(z) = \max\{\tau, v_{-j}^\mathsf{T} o_j + c\} = \max\{\tau - c, v_{-j}^\mathsf{T} o_j\} + c$. Note, $o_j = \sum_{k=1}^{n} z_{j,k}$ represents the input source information of the $j^{\text{th}}$ output and $v_{-j} = 2\sum_{j'=1, j'\neq j}^{m'} A o_{j'}$ encodes the status of the other outputs. Thus, we can interpret the supermodular term as a penalization of each label of $o_j$ in proportion to the corresponding $v_{-j}$ value (criterion 1), but not for the compatibility below $\tau - c$ (criterion 2). As a modular function which satisfies the criteria above, we use the following function:

$$f_c(z) \approx \max\{\tau', v_{-j}\}^\mathsf{T} o_j \quad \text{for } \exists \tau' \in \mathbb{R}. \tag{2}$$

Note that, by satisfying the criteria above, the modular function reflects the diversity and over-penalization desiderata described in Section 4.1. We illustrate the proposed mixup procedure with the modularized diversity penalty in Figure 3.

**Proposition 3.** *The modularization given by Equation* (2) *satisfies the criteria above.*

*Proof.* See Appendix B.2. □

By applying the modular approximation described in Equation (2) to $f_c$ in Equation (1), we can iteratively apply a submodular minimization algorithm to obtain the final solution as described in Algorithm 1. In detail, each step can be performed as follows: 1) Conditioning the main objective $f$ on the current values except $z_j$, denoted as $f_j(z_j) = f(z_j; z_{1:j-1}, z_{j+1:m'})$. 2) Modularization of the compatibility term of $f_j$ as Equation (2), resulting in a submodular function $\tilde{f}_j$. We denote the modularization operator as $\Phi$, i.e., $\tilde{f}_j = \Phi(f_j)$. 3) Applying a submodular minimization algorithm to $\tilde{f}_j$. Please refer to Appendix C for implementation details.

---

**Algorithm 1** Iterative submodular minimization

Initialize $z$ as $z^{(0)}$.
Let $z^{(t)}$ denote a solution of the $t^{\text{th}}$ step.
$\Phi$: modularization operator based on Equation (2).
**for** $t = 1, \ldots, T$ **do**
  **for** $j = 1, \ldots, m'$ **do**
    $f_j^{(t)}(z_j) := f(z_j; z_{1:j-1}^{(t)}, z_{j+1:m'}^{(t-1)})$.
    $\tilde{f}_j^{(t)} = \Phi(f_j^{(t)})$.
    Solve $z_j^{(t)} = \arg\min \tilde{f}_j^{(t)}(z_j)$.
  **end for**
**end for**
**return** $z^{(T)}$

---

**Analysis** Narasimhan and Bilmes (2005) proposed a modularization strategy for general supermodular set functions, and apply a submodular minimization algorithm that can monotonically decrease the original BP objective. However, the proposed Algorithm 1 based on Equation (2) is much more suitable for minibatch based mixup for neural network training than the set modularization proposed by Narasimhan and Bilmes (2005) in terms of complexity and modularization variance due to randomness. For simplicity, let us assume

each $z_{j,k}$ is an $m$-dimensional one-hot vector. Then, our problem is to optimize $m'n$ one-hot $m$-dimensional vectors.

To apply the set modularization method, we need to assign each possible value of $z_{j,k}$ as an element of $\{1, 2, \ldots, m\}$. Then the supermodular term in Equation (1) can be interpreted as a set function with $m'nm$ elements, and to apply the set modularization, $O(m'nm)$ sequential evaluations of the supermodular term are required. In contrast, Algorithm 1 calculates $v_{\cdot j}$ in Equation (2) in only $O(m')$ time per each iteration. In addition, each modularization step of the set modularization method requires a random permutation of the $m'nm$ elements. In this case, the optimization can be strongly affected by the randomness from the permutation step. As a result, the optimal labeling of each $z_{j,k}$ from the compatibility term is strongly influenced by the random ordering undermining the interpretability of the algorithm. Please refer to Appendix D for empirical comparison between Algorithm 1 and the method by Narasimhan and Bilmes (2005).

## 5 EXPERIMENTS

We evaluate our proposed mixup method on generalization, weakly supervised object localization, calibration, and robustness tasks. First, we compare the generalization performance of the proposed method against baselines by training classifiers on CIFAR-100 (Krizhevsky et al., 2009), Tiny-ImageNet (Chrabaszcz et al., 2017), ImageNet (Deng et al., 2009), and the Google commands speech dataset (Warden, 2017). Next, we test the localization performance of classifiers following the evaluation protocol of Qin and Kim (2019). We also measure calibration error (Guo et al., 2017) of classifiers to verify Co-Mixup successfully alleviates the over-confidence issue by Zhang et al. (2018). In Section 5.4, we evaluate the robustness of the classifiers on the test dataset with background corruption in response to the recent problem raised by Lee et al. (2020) that deep neural network agents often fail to generalize to unseen environments. Finally, we perform a sensitivity analysis of Co-Mixup and provide the results in Appendix F.3.

### 5.1 CLASSIFICATION

We first train PreActResNet18 (He et al., 2016), WRN16-8 (Zagoruyko and Komodakis, 2016), and ResNeXt29-4-24 (Xie et al., 2017) on CIFAR-100 for 300 epochs. We use stochastic gradient descent with an initial learning rate of 0.2 decayed by factor 0.1 at epochs 100 and 200. We set the momentum as 0.9 and add a weight decay of 0.0001. With this setup, we train a vanilla classifier and reproduce the mixup baselines (Zhang et al., 2018; Verma et al., 2019; Yun et al., 2019; Kim et al., 2020), which we denote as *Vanilla, Input, Manifold, CutMix, Puzzle Mix* in the experiment tables. Note that we use identical hyperparameters regarding Co-Mixup over all of the experiments with different models and datasets, which are provided in Appendix E.

Table 1 shows Co-Mixup significantly outperforms all other baselines in Top-1 error rate. Co-Mixup achieves 19.87% in Top-1 error rate with PreActResNet18, outperforming the best baseline by 0.75%. We further test Co-Mixup on different models (WRN16-8 & ResNeXt29-4-24) and verify Co-Mixup improves Top-1 error rate over the best performing baseline.

| Dataset (Model) | Vanilla | Input | Manifold | CutMix | Puzzle Mix | Co-Mixup |
|---|---|---|---|---|---|---|
| CIFAR-100 (PreActResNet18) | 23.59 | 22.43 | 21.64 | 21.29 | 20.62 | **19.87** |
| CIFAR-100 (WRN16-8) | 21.70 | 20.08 | 20.55 | 20.14 | 19.24 | **19.15** |
| CIFAR-100 (ResNeXt29-4-24) | 21.79 | 21.70 | 22.28 | 21.86 | 21.12 | **19.78** |
| Tiny-ImageNet (PreActResNet18) | 43.40 | 43.48 | 40.76 | 43.11 | 36.52 | **35.85** |
| ImageNet (ResNet-50, 100 epochs) | 24.03 | 22.97 | 23.30 | 22.92 | 22.49 | **22.39** |
| Google commands (VGG-11) | 4.84 | 3.91 | 3.67 | 3.76 | 3.70 | **3.54** |

Table 1: Top-1 error rate on various datasets and models. For CIFAR-100, we train each model with three different random seeds and report the mean error.

We further test Co-Mixup on other datasets; Tiny-ImageNet, ImageNet, and the Google commands dataset (Table 1). For Tiny-ImageNet, we train PreActResNet18 for 1200 epochs

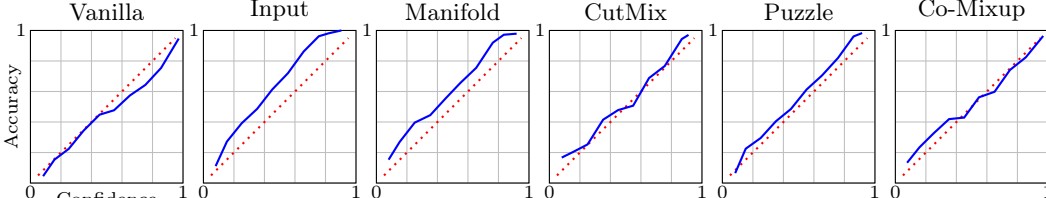

Figure 4: Confidence-Accuracy plots for classifiers on CIFAR-100. From the figure, the Vanilla network shows over-confident predictions, whereas other mixup baselines tend to have under-confident predictions. We can find that Co-Mixup has best-calibrated predictions.

following the training protocol of Kim et al. (2020). As a result, Co-Mixup consistently improves Top-1 error rate over baselines by 0.67%. In the ImageNet experiment, we follow the experimental protocol provided in Puzzle Mix (Kim et al., 2020), which trains ResNet-50 (He et al., 2015) for 100 epochs. As a result, Co-Mixup outperforms all of the baselines in Top-1 error rate. We further test Co-Mixup on the speech domain with the Google commands dataset and VGG-11 (Simonyan and Zisserman, 2014). We provide a detailed experimental setting and dataset description in Appendix F.1. From Table 1, we confirm that Co-Mixup is the most effective in the speech domain as well.

## 5.2    Localization

We compare weakly supervised object localization (WSOL) performance of classifiers trained on ImageNet (in Table 1) to demonstrate that our mixup method better guides a classifier to focus on salient regions. We test the localization performance using CAM (Zhou et al., 2016), a WSOL method using a pre-trained classifier. We evaluate localization performance following the evaluation protocol in Qin and Kim (2019), with binarization threshold 0.25 in CAM. Table 2 summarizes the WSOL performance of various mixup methods, which shows that our proposed mixup method outperforms other baselines.

## 5.3    Calibration

We evaluate the expected calibration error (ECE) (Guo et al., 2017) of classifiers trained on CIFAR-100. Note, ECE is calculated by the weighted average of the absolute difference between the confidence and accuracy of a classifier. As shown in Table 2, the Co-Mixup classifier has the lowest calibration error among baselines. From Figure 4, we find that other mixup baselines tend to have *under-confident* predictions resulting in higher ECE values even than *Vanilla* network (also pointed out by Wen et al. (2020)), whereas Co-Mixup has best-calibrated predictions resulting in relatively 48% less ECE value. We provide more figures and results with other datasets in Appendix F.2.

| Task | Vanilla | Input | Manifold | CutMix | Puzzle Mix | Co-Mixup |
|---|---|---|---|---|---|---|
| Localization (Acc. %) (↑) | 54.36 | 55.07 | 54.86 | 54.91 | 55.22 | **55.32** |
| Calibration   (ECE %) (↓) | 3.9 | 17.7 | 13.1 | 5.6 | 7.5 | **1.9** |

Table 2: WSOL results on ImageNet and ECE (%) measurements of CIFAR-100 classifiers.

## 5.4    Robustness

In response to the recent problem raised by Lee et al. (2020) that deep neural network agents often fail to generalize to unseen environments, we consider the situation where the statistics of the foreground object, such as color or shape, is unchanged, but with the corrupted (or replaced) background. In detail, we consider the following operations: 1) replacement with another image and 2) adding Gaussian noise. We use ground-truth bounding boxes to separate the foreground from the background, and then apply the previous operations independently to obtain test datasets. We provide a detailed description of datasets in Appendix G.

With the test datasets described above, we evaluate the robustness of the pre-trained classifiers. As shown in Table 3, Co-Mixup shows significant performance gains at various background corruption tests compared to the other mixup baselines. For each corruption case, the classifier trained with Co-Mixup outperforms the others in Top-1 error rate with the performance margins of 2.86% and 3.33% over the Vanilla model.

| Corruption type | Vanilla | Input | Manifold | CutMix | Puzzle Mix | Co-Mixup |
|---|---|---|---|---|---|---|
| Random replacement | 41.63 | 39.41 | 39.72 | 46.20 | 39.23 | **38.77** |
| | (+17.62) | (+16.47) | (+16.47) | (+23.16) | (+16.69) | (+16.38) |
| Gaussian noise | 29.22 | 26.29 | 26.79 | 27.13 | 26.11 | **25.89** |
| | (+5.21) | (+3.35) | (+3.54) | (+4.09) | (+3.57) | (+3.49) |

Table 3: Top-1 error rates of various mixup methods for background corrupted ImageNet validation set. The values in the parentheses indicate the error rate increment by corrupted inputs compared to clean inputs.

## 5.5 Baselines with multiple inputs

To further investigate the effect of the number of inputs for the mixup in isolation, we conduct an ablation study on baselines using multiple mixing inputs. For fair comparison, we use Dirichlet$(\alpha, \ldots, \alpha)$ prior for the mixing ratio distribution and select the best performing $\alpha$ in $\{0.2, 1.0, 2.0\}$. Note that we overlay multiple boxes in the case of CutMix. Table 4 reports the classification test errors on CIFAR-100 with PreActResNet18. From the table, we find that mixing multiple inputs decreases the performance gains of each mixup baseline. These results demonstrate that mixing multiple inputs could lead to possible degradation of the performance and support the necessity of considering saliency information and diversity as in Co-Mixup.

| # inputs for mixup | Input | Manifold | CutMix | Co-Mixup |
|---|---|---|---|---|
| # inputs = 2 | 22.43 | 21.64 | 21.29 | |
| # inputs = 3 | 23.03 | 22.13 | 22.01 | 19.87 |
| # inputs = 4 | 23.12 | 22.07 | 22.20 | |

Table 4: Top-1 error rates of mixup baselines with multiple mixing inputs on CIFAR-100 and PreActResNet18. We report the mean values of three different random seeds. Note that Co-Mixup optimally determines the number of inputs for each output by solving the optimization problem.

## 6 Conclusion

We presented Co-Mixup for optimal construction of a batch of mixup examples by finding the best combination of salient regions among a collection of input data while encouraging diversity among the generated mixup examples. This leads to a discrete optimization problem minimizing a novel submodular-supermodular objective. In this respect, we present a practical modular approximation and iterative submodular optimization algorithm suitable for minibatch based neural network training. Our experiments on generalization, weakly supervised object localization, and robustness against background corruption show Co-Mixup achieves the state of the art performance compared to other mixup baseline methods. The proposed generalized mixup framework tackles the important question of 'what to mix?' while the existing methods only consider 'how to mix?'. We believe this work can be applied to new applications where the existing mixup methods have not been applied, such as multi-label classification, multi-object detection, or source separation.

## Acknowledgements

This research was supported in part by Samsung Advanced Institute of Technology, Samsung Electronics Co., Ltd, Institute of Information & Communications Technology Planning & Evaluation (IITP) grant funded by the Korea government (MSIT) (No. 2020-0-00882, (SW STAR LAB) Development of deployable learning intelligence via self-sustainable and trustworthy machine learning), and Research Resettlement Fund for the new faculty of Seoul National University. Hyun Oh Song is the corresponding author.

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

# A    SUPPLEMENTARY NOTES FOR OBJECTIVE

## A.1    NOTATIONS

In Table 5, we provide a summary of notations in the main text.

| Notation | Meaning |
|---|---|
| $m$, $m'$, $n$ | # inputs, # outputs, dimension of data |
| $c_k \in \mathbb{R}^m$ $(1 \le k \le n)$ | labeling cost for $m$ input sources at the $k^{th}$ location |
| $z_{j,k} \in \mathcal{L}^m$ $(1 \le j \le m', 1 \le k \le n)$ | input source ratio at the $k^{th}$ location of the $j^{th}$ output |
| $z_j \in \mathcal{L}^{m \times n}$ | labeling of the $j^{th}$ output |
| $o_j \in \mathbb{R}^m$ | aggregation of the labeling of the $j^{th}$ output |
| $A \in \mathbb{R}^{m \times m}$ | compatibility between inputs |

Table 5: A summary of notations.

## A.2    INTERPRETATION OF COMPATIBILITY

In our main objective Equation (1), we introduce a compatibility matrix $A = (1-\omega)I + \omega A_c$ between inputs. By minimizing $\langle o_j, o_{j'} \rangle_A$ for $j \ne j'$, we encourage each individual mixup examples to have high compatibility within. Figure 5 explains how the compatibility term works by comparing simple cases. Note that our framework can reflect any compatibility measures for the optimal mixup.

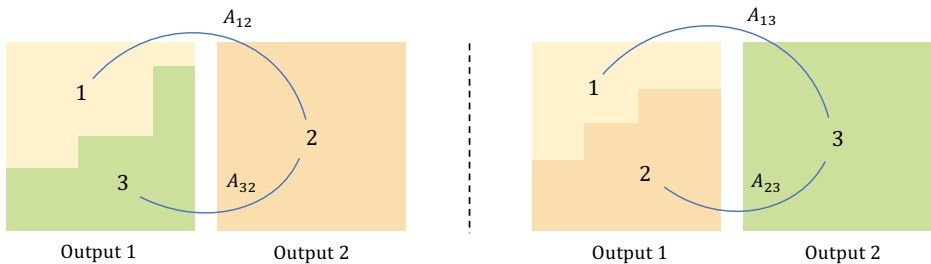

Figure 5: Let us consider Co-Mixup with three inputs and two outputs. The figure represents two Co-Mixup results. Each input is denoted as a number and color-coded. Let us assume that input 1 and input 2 are more compatible, *i.e.*, $A_{12} \gg A_{23}$ and $A_{12} \gg A_{13}$. Then, the left Co-Mixup result has a larger inner-product value $\langle o_1, o_2 \rangle_A$ than the right. Thus the mixup result on the right has higher compatibility than the result on the left within each output example.

# B    PROOFS

## B.1    PROOF OF PROPOSITION 1

**Lemma 1.** *For a positive semi-definite matrix $A \in \mathbb{R}_+^{m \times m}$ and $x, x' \in \mathbb{R}^m$, $s(x, x') = x^\intercal A x'$ is pairwise supermodular.*

*Proof.* $s(x, x) + s(x', x') - s(x, x') - s(x', x) = x^\intercal A x + x^\intercal A x - 2x^\intercal A x' = (x-x')^\intercal A(x-x')$, and because $A$ is positive semi-definite, $(x-x')^\intercal A(x-x') \ge 0$. $\square$

**Proposition 1.** *The compatibility term $f_c$ in Equation (1) is pairwise supermodular for every pair of $(z_{j_1,k}, z_{j_2,k})$ if $A$ is positive semi-definite.*

*Proof.* For $j_1$ and $j_2$, *s.t.*, $j_1 \neq j_2$, $\max \left\{ \tau, \sum_{j=1}^{m'} \sum_{j'=1,j'\neq j}^{m'} (\sum_{k=1}^{n} z_{j,k})^{\mathsf{T}} A (\sum_{k=1}^{n} z_{j',k}) \right\} = \max\{\tau, c + 2z_{j_1,k}^{\mathsf{T}} A z_{j_2,k}\} = -\min\{-\tau, -c - 2z_{j_1,k}^{\mathsf{T}} A z_{j_2,k}\}$, for $\exists c \in \mathbb{R}$. By Lemma 1, $-z_{j_1,k}^{\mathsf{T}} A z_{j_2,k}$ is pairwise submodular, and because a budget additive function preserves submodularity (Horel and Singer, 2016), $\min\{-\tau, -c - 2z_{j_1,k}^{\mathsf{T}} A z_{j_2,k}\}$ is pairwise submodular with respect to $(z_{j_1,k}, z_{j_2,k})$. $\square$

### B.2 Proof of proposition 3

**Proposition 3.** *The modularization given by Equation (2) satisfies the criteria.*

*Proof.* Note, by the definition in Equation (1), the compatibility between the $j^{th}$ and $j'^{th}$ outputs is $o_{j'}^{\mathsf{T}} A o_j$, and thus, $v_{-j}^{\mathsf{T}} o_j$ represents the compatibility between the $j^{th}$ output and the others. In addition, $\|o_j\|_1 = \|\sum_{k=1}^{n} z_{j,k}\|_1 = \sum_{k=1}^{n} \|z_{j,k}\|_1 = n$. In a local view, for the given $o_j$, let us define a vector $o_j'$ as $o_j'[i_1] = o_j[i_1] + \alpha$ and $o_j'[i_2] = o_j[i_2] - \alpha$ for $\alpha > 0$. Without loss of generality, let us assume $v_{-j}$ is sorted in ascending order. Then, $v_{-j}^{\mathsf{T}} o_j \leq v_{-j}^{\mathsf{T}} o_j'$ implies $i_1 > i_2$, and because the max function preserves the ordering, $\max\{\tau', v_{-j}\}^{\mathsf{T}} o_j \leq \max\{\tau', v_{-j}\}^{\mathsf{T}} o_j'$. Thus, the criterion 1 is locally satisfied. Next, for $\tau' > 0$, $\|\max\{\tau', v_{-j}\}^{\mathsf{T}} o_j\|_1 \geq \tau' \|o_j\|_1 = \tau' n$. Let $\exists i_0$ s.t. for $i < i_0, v_{-j}[i] < \tau'$, and for $i \geq i_0, v_{-j}[i] \geq \tau'$. Then, for $o_j$ containing positive elements only in indices smaller than $i_0$, $\max\{\tau', v_{-j}\}^{\mathsf{T}} o_j = \tau' n$ which means there is no extra penalty from the compatibility. In this respect, the proposed modularization satisfies the criterion 2 as well. $\square$

## C Implementation details

We perform the optimization after down-sampling the given inputs and saliency maps to the specified size ($4 \times 4$). After the optimization, we up-sample the optimal labeling to match the size of the inputs and then mix inputs according to the up-sampled labeling. For the saliency measure, we calculate the gradient values of training loss with respect to the input data and measure $\ell_2$ norm of the gradient values across input channels (Simonyan et al., 2013). In classification experiments, we retain the gradient information of network weights obtained from the saliency calculation for regularization. For the distance in the compatibility term, we measure $\ell_1$-distance between the most salient regions.

For the initialization in Algorithm 1, we use *i.i.d.* samples from a categorical distribution with equal probabilities. We use *alpha-beta* swap algorithm from pyGCO[1] to solve the minimization step in Algorithm 1, which can find local-minima of a multi-label submodular function. However, the worst-case complexity of *alpha-beta* swap algorithm with $|\mathcal{L}| = 2$ is $O(m^2 n)$, and in the case of mini-batch with 100 examples, iteratively applying the algorithm can become a bottleneck during the network training. To mitigate the computational overhead, we partition the mini-batch (each of size 20) and then apply Algorithm 1 independently per each partition.

The worst-case complexity theoretic of the naive implementation of Algorithm 1 increases exponentially as $|\mathcal{L}|$ increases. Specifically, the worst-case theoretic complexity of the *alpha-beta* swap algorithm is proportional to the square of the number of possible states of $z_{j,k}$, which is proportional to $m^{|\mathcal{L}|-1}$. To reduce the number of possible states in a multi-label case, we solve the problem for binary labels ($|\mathcal{L}| = 2$) at the first inner-cycle and then extend to multi labels ($|\mathcal{L}| = 3$) only for the currently assigned indices of each output in the subsequent cycles. This reduces the number of possible states to $O(m + \bar{m}^{|\mathcal{L}|-1})$ where $\bar{m} \ll m$. Here, $\bar{m}$ means the number of currently assigned indices for each output.

---

[1] https://github.com/Borda/pyGCO

Based on the above implementation, we train models with Co-Mixup in a feasible time. For example, in the case of ImageNet training with 16 Intel I9-9980XE CPU cores and 4 NVIDIA RTX 2080Ti GPUs, Co-Mixup training requires 0.964s per batch, whereas the vanilla training without mixup requires 0.374s per batch. Note that Co-Mixup requires saliency computation, and when we compare the algorithm with Puzzle Mix, which performs the same saliency computation, Co-Mixup is only slower about 1.04 times. Besides, as we down-sample the data to the fixed size regardless of the data dimension, the additional computation cost of Co-Mixup relatively decreases as the data dimension increases. Finally, we present the empirical time complexity of Algorithm 1 in Figure 6. As shown in the figure, Algorithm 1 has linear time complexity over $|\mathcal{L}|$ empirically. Note that we use $|\mathcal{L}| = 3$ in all of our main experiments, including a classification task.

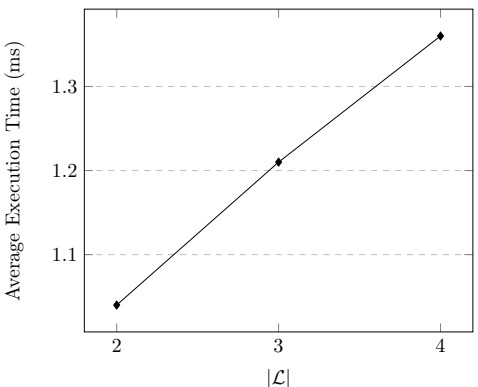 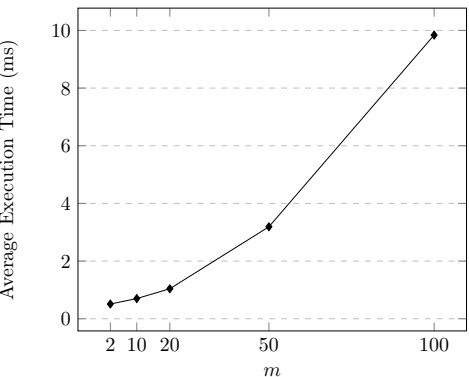

Figure 6: Mean execution time (ms) of Algorithm 1 per each batch of data over 100 trials. The left figure shows the time complexity of the algorithm over $|\mathcal{L}|$ and the right figure shows the time complexity over the number of inputs $m$. Note that the other parameters are fixed equal to the classification experiments setting, $m = m' = 20$, $n = 16$, and $|\mathcal{L}| = 3$.

## D ALGORITHM ANALYSIS

In this section, we perform comparison experiments to analyze the proposed Algorithm 1. First, we compare our algorithm with the exact brute force search algorithm to inspect the optimality of the algorithm. Next, we compare our algorithm with the BP algorithm proposed by Narasimhan and Bilmes (2005).

### D.1 COMPARISON WITH BRUTE FORCE

To inspect the optimality of the proposed algorithm, we compare the function values of the solutions of Algorithm 1, brute force search algorithm, and random guess. Due to the exponential time complexity of the brute force search, we compare the algorithms on small scale experiment settings. Specifically, we test algorithms on settings of $(m = m' = 2, \ n = 4)$, $(m = m' = 2, \ n = 9)$, and $(m = m' = 3, \ n = 4)$ varying the number of inputs and outputs $(m, \ m')$ and the dimension of data $n$. We generate unary cost matrix in the objective $f$ by sampling data from uniform distribution.

We perform experiments with 100 different random seeds and summarize the results on Table 6. From the table, we find that the proposed algorithm achieves near optimal solutions over various settings. We also measure relative errors between ours and random guess, $(f(z_{\text{ours}}) - f(z_{\text{brute}}))/(f(z_{\text{random}}) - f(z_{\text{brute}}))$. As a result, our algorithm achieves relative error less than 0.01.

### D.2 COMPARISON WITH ANOTHER BP ALGORITHM

We compare the proposed algorithm and the BP algorithm proposed by Narasimhan and Bilmes (2005). We evaluate function values of solutions by each method using a random

| Configuration | Ours | Brute force (optimal) | Random guess | Rel. error |
|---|---|---|---|---|
| $(m = m' = 2, \ n = 4)$ | 1.91 | 1.90 | 3.54 | 0.004 |
| $(m = m' = 2, \ n = 9)$ | 1.93 | 1.91 | 3.66 | 0.01 |
| $(m = m' = 3, \ n = 4)$ | 2.89 | 2.85 | 22.02 | 0.002 |

Table 6: Mean function values of the solutions over 100 different random seeds. Rel. error means the relative error between ours and random guess.

unary cost matrix from a uniform distribution. We compare methods over various scales by controlling the number of mixing inputs $m$.

Table 7 shows the averaged function values with standard deviations in the parenthesis. As we can see from the table, the proposed algorithm achieves much lower function values and deviations than the method by Narasimhan and Bilmes (2005) over various settings. Note that the method by Narasimhan and Bilmes (2005) has high variance due to randomization in the algorithm. We further compare the algorithm convergence time in Table 8. The experiments verify that the proposed algorithm is much faster and effective than the method by Narasimhan and Bilmes (2005).

| Algorithm | $m = 5$ | $m = 10$ | $m = 20$ | $m = 50$ | $m = 100$ |
|---|---|---|---|---|---|
| Ours | 3.1 (1.7) | 15 (6.6) | 54 (15) | 205 (26) | 469 (31) |
| Narasimhan | 269 (58) | 1071 (174) | 4344 (701) | 24955 (4439) | 85782 (14337) |
| Random | 809 (22) | 7269 (92) | 60964 (413) | 980973 (2462) | 7925650 (10381) |

Table 7: Mean function values of the solutions over 100 different random seeds. We report the standard deviations in the parenthesis. Random represents the random guess algorithm.

| Algorithm | $m = 5$ | $m = 10$ | $m = 20$ | $m = 50$ | $m = 100$ |
|---|---|---|---|---|---|
| Ours | 0.02 | 0.04 | 0.11 | 0.54 | 2.71 |
| Narasimhan | 0.06 | 0.09 | 0.27 | 1.27 | 7.08 |

Table 8: Convergence time (s) of the algorithms.

## E    Hyperparameter settings

We perform Co-Mixup after down-sampling the given inputs and saliency maps to the pre-defined resolutions regardless of the size of the input data. In addition, we normalize the saliency of each input to sum up to 1 and define unary cost using the normalized saliency. As a result, we use an identical hyperparameter setting for various datasets; CIFAR-100, Tiny-ImageNet, and ImageNet. In details, we use $(\beta, \gamma, \eta, \tau) = (0.32, 1.0, 0.05, 0.83)$ for all of experiments. Note that $\tau$ is normalized according to the size of inputs $(n)$ and the ratio of the number of inputs and outputs $(m/m')$, and we use an isotropic Dirichlet distribution with $\alpha = 2$ for prior $p$. For a compatibility matrix, we use $\omega = 0.001$.

For baselines, we tune the mixing ratio hyperparameter, *i.e.,* the beta distribution parameter (Zhang et al., 2018), among $\{0.2, 1.0, 2.0\}$ for all of the experiments if the specific setting is not provided in the original papers.

## F    Additional Experimental Results

### F.1    Another Domain: Speech

In addition to the image domain, we conduct experiments on the speech domain, verifying Co-Mixup works on various domains. Following (Zhang et al., 2018), we train LeNet (LeCun

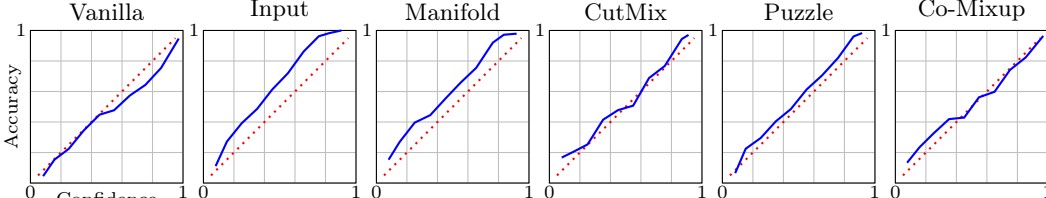

Figure 7: Confidence-Accuracy plots for classifiers on CIFAR-100. Note, ECE is calculated by the mean absolute difference between the two values.

et al., 1998) and VGG-11 (Simonyan and Zisserman, 2014) on the Google commands dataset (Warden, 2017). The dataset consists of 65,000 utterances, and each utterance is about one-second-long belonging to one out of 30 classes. We train each classifier for 30 epochs with the same training setting and data pre-processing of Zhang et al. (2018). In more detail, we use $160 \times 100$ normalized spectrograms of utterances for training. As shown in Table 9, we verify that Co-Mixup is still effective in the speech domain.

| Model | Vanilla | Input | Manifold | CutMix | Puzzle Mix | Co-Mixup |
|---|---|---|---|---|---|---|
| LeNet | 11.24 | 10.83 | 12.33 | 12.80 | 10.89 | **10.67** |
| VGG-11 | 4.84 | 3.91 | 3.67 | 3.76 | 3.70 | **3.57** |

Table 9: Top-1 classification test error on the Google commands dataset. We stop training if validation accuracy does not increase for 5 consecutive epochs.

## F.2 CALIBRATION

In this section, we summarize the expected calibration error (ECE) (Guo et al., 2017) of classifiers trained with various mixup methods. For evaluation, we use the official code provided by the TensorFlow-Probability library[2] and set the number of bins as 10. As shown in Table 10, Co-Mixup classifiers have the lowest calibration error on CIFAR-100 and Tiny-ImageNet. As pointed by Guo et al. (2017), the Vanilla networks have over-confident predictions, but however, we find that mixup classifiers tend to have under-confident predictions (Figure 7; Figure 8). As shown in the figures, Co-Mixup successfully alleviates the over-confidence issue and does not suffer from under-confidence predictions.

| Dataset | Vanilla | Input | Manifold | CutMix | Puzzle Mix | Co-Mixup |
|---|---|---|---|---|---|---|
| CIFAR-100 | 3.9 | 17.7 | 13.1 | 5.6 | 7.5 | **1.9** |
| Tiny-ImageNet | 4.5 | 6.2 | 6.8 | 12.0 | 5.6 | **2.5** |
| ImageNet | 5.9 | **1.2** | 1.7 | 4.3 | 2.1 | 2.1 |

Table 10: Expected calibration error (%) of classifiers trained with various mixup methods on CIFAR-100, Tiny-ImageNet and ImageNet. Note that, at all of three datasets, Co-Mixup outperforms all of the baselines in Top-1 accuracy.

## F.3 SENSITIVITY ANALYSIS

We measure the Top-1 error rate of the model by sweeping the hyperparameter to show the sensitivity using PreActResNet18 on CIFAR-100 dataset. We sweep the label smoothness coefficient $\beta \in \{0, 0.16, 0.32, 0.48, 0.64\}$, compatibility coefficient $\gamma \in \{0.6, 0.8, 1.0, 1.2, 1.4\}$, clipping level $\tau \in \{0.79, 0.81, 0.83, 0.85, 0.87\}$, compatibility matrix parameter $\omega \in \{0, 5 \cdot 10^{-4}, 10^{-3}, 5 \cdot 10^{-3}, 10^{-2}\}$, and the size of partition $m \in \{2, 4, 10, 20, 50\}$. Table 11 shows that Co-Mixup outperforms the best baseline (PuzzleMix, 20.62%) with a large pool of

---

[2]https://www.tensorflow.org/probability/api_docs/python/tfp/stats/expected_calibration_error

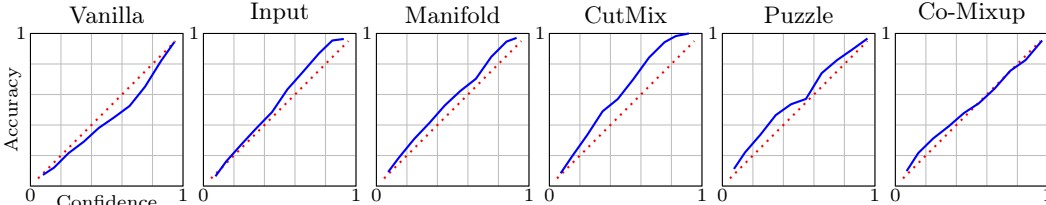

Figure 8: Confidence-Accuracy plots for classifiers on Tiny-ImageNet.

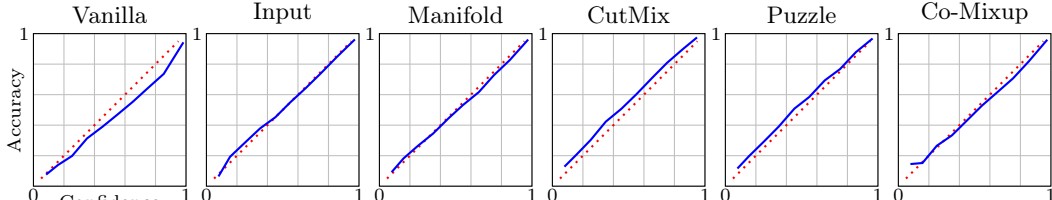

Figure 9: Confidence-Accuracy plots for classifiers on ImageNet.

hyperparameters. We also find that Top-1 error rate increases as the partition batch size $m$ increases until $m = 20$.

| Smoothness coefficient, $\beta$ | $\beta = 0$ | $\beta = 0.16$ | $\beta = 0.32$ | $\beta = 0.48$ | $\beta = 0.64$ |
|---|---|---|---|---|---|
| | 20.29 | 20.18 | **19.87** | 20.35 | 21.24 |
| Compatibility coefficient, $\gamma$ | $\gamma = 0.6$ | $\gamma = 0.8$ | $\gamma = 1.0$ | $\gamma = 1.2$ | $\gamma = 1.4$ |
| | 20.3 | 19.99 | **19.87** | 20.09 | 20.13 |
| Clipping parameter, $\tau$ | $\tau = 0.79$ | $\tau = 0.81$ | $\tau = 0.83$ | $\tau = 0.85$ | $\tau = 0.87$ |
| | 20.45 | 20.14 | **19.87** | 20.15 | 20.23 |
| Compatibility matrix parameter, $\omega$ | $\omega = 0$ | $\omega = 5 \cdot 10^{-4}$ | $\omega = 10^{-3}$ | $\omega = 5 \cdot 10^{-3}$ | $\omega = 10^{-2}$ |
| | 20.51 | 20.42 | **19.87** | 20.18 | 20.14 |
| Partition size, $m$ | $m = 2$ | $m = 4$ | $m = 10$ | $m = 20$ | $m = 50$ |
| | 20.3 | 20.22 | 20.15 | **19.87** | 19.96 |

Table 11: Hyperparameter sensitivity results (Top-1 error rates) on CIFAR-100 with PreActResNet18. We report the mean values of three different random seeds.

### F.4 COMPARISON WITH NON-MIXUP BASELINES

We compare the generalization performance of Co-Mixup with non-mixup baselines, verifying the proposed method achieves the state of the art generalization performance not only for the mixup-based methods but for other general regularization based methods. One of the regularization methods called VAT (Miyato et al., 2018) uses virtual adversarial loss, which is defined as the KL-divergence of predictions between input data against local perturbation. We perform the experiment with VAT regularization on CIFAR-100 with PreActResNet18 for 300 epochs in the supervised setting. We tune $\alpha$ (coefficient of VAT regularization term) in $\{0.001, 0.01, 0.1\}$, $\epsilon$ (radius of $\ell$-inf ball) in $\{1, 2\}$, and the number of noise update steps in $\{0, 1\}$. Table 12 shows that Co-Mixup, which achieves Top-1 error rate of 19.87%, outperforms the VAT regularization method.

## G    DETAILED DESCRIPTION FOR BACKGROUND CORRUPTION

We build the background corrupted test datasets based on ImageNet validation dataset to compare the robustness of the pre-trained classifiers against the background corruption.

| VAT loss coefficient | # update=0 | | # update=1 | |
|---|---|---|---|---|
| | $\epsilon = 1$ | $\epsilon = 2$ | $\epsilon = 1$ | $\epsilon = 2$ |
| $\alpha = 0.001$ | 23.38 | 23.62 | 24.76 | 26.22 |
| $\alpha = 0.01$ | 23.14 | 23.67 | 28.33 | 31.95 |
| $\alpha = 0.1$ | 23.65 | 23.88 | 34.75 | 39.82 |

Table 12: Top-1 error rates of VAT on CIFAR-100 dataset with PreActResNet18.

ImageNet consists of images $\{x_1, ..., x_M\}$, labels $\{y_1, ..., y_M\}$, and the corresponding ground-truth bounding boxes $\{b_1, ..., b_M\}$. We use the ground-truth bounding boxes to separate the foreground from the background. Let $z_j$ be a binary mask of image $x_j$, which has value 1 inside of the ground-truth bounding box $b_j$. Then, we generate two types of background corrupted sample $\tilde{x}_j$ by considering the following operations:

1. Replacement with another image as $\tilde{x}_j = x_j \odot z_j + x_{i(j)} \odot (1 - z_j)$ for a random permutation $\{i(1), ..., i(M)\}$.

2. Adding Gaussian noise as $\tilde{x}_j = x_j \odot z_j + \epsilon \odot (1 - z_j)$, where $\epsilon \sim N(0, 0.1^2)$. We clip pixel values of $\tilde{x}_j$ to [0, 1].

Figure 10 visualizes subsets of the background corruption test datasets.

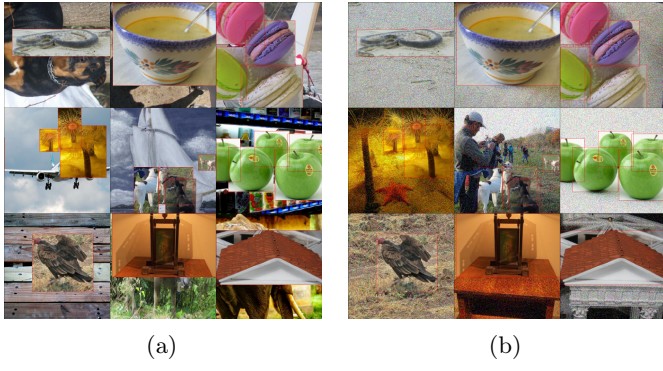

(a)           (b)

Figure 10: Each subfigure shows background corrupted samples used in the robustness experiment. (a) Replacement with another image in ImageNet. (b) Adding Gaussian noise. The red boxes on the images represent ground-truth bounding boxes.

## H  CO-MIXUP GENERATED SAMPLES

In Figure 12, we present Co-Mixup generated image samples by using images from ImageNet. We use an input batch consisting of 24 images, which is visualized in Figure 11. As can be seen from Figure 12, Co-Mixup efficiently mix-matches salient regions of the given inputs maximizing saliency and creates diverse outputs. In Figure 12, inputs with the target objects on the left side are mixed with the objects on the right side, and objects on the top side are mixed with the objects on the bottom side. In Figure 13, we present Co-Mixup generated image samples with larger $\tau$ using the same input batch. By increasing $\tau$, we can encourage Co-Mixup to use more inputs to mix per each output.

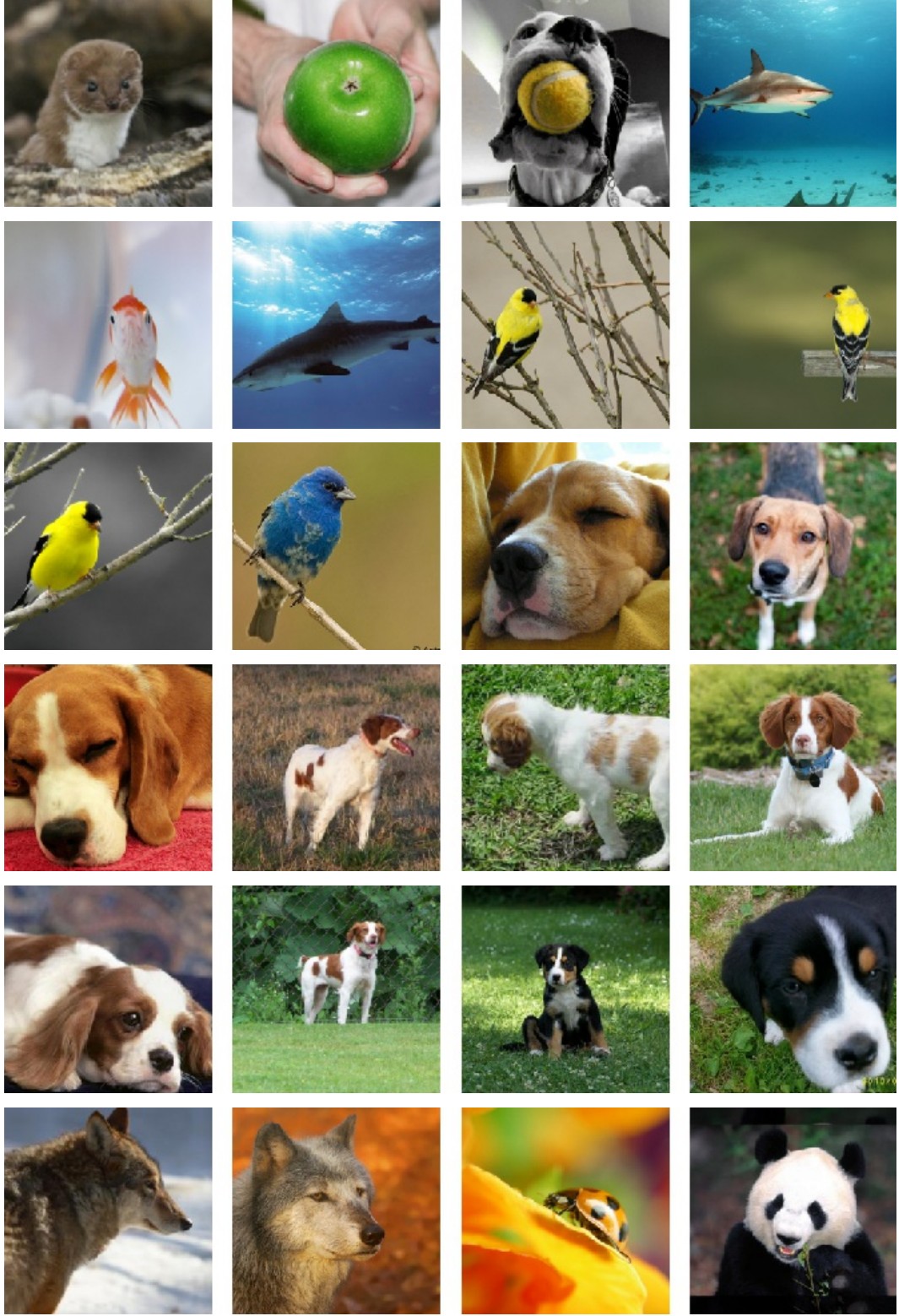

Figure 11: Input batch.

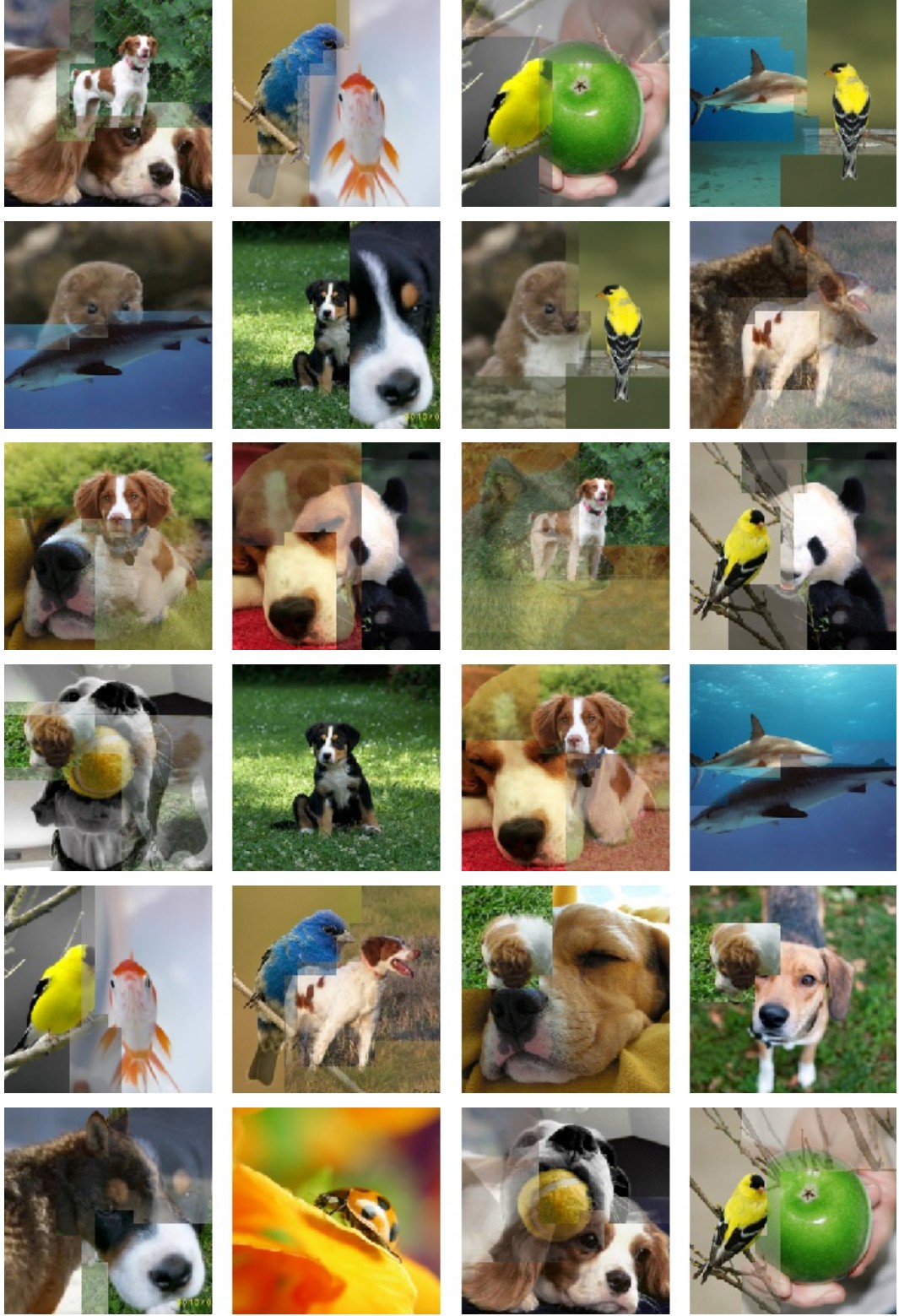

Figure 12: Mixed output batch.

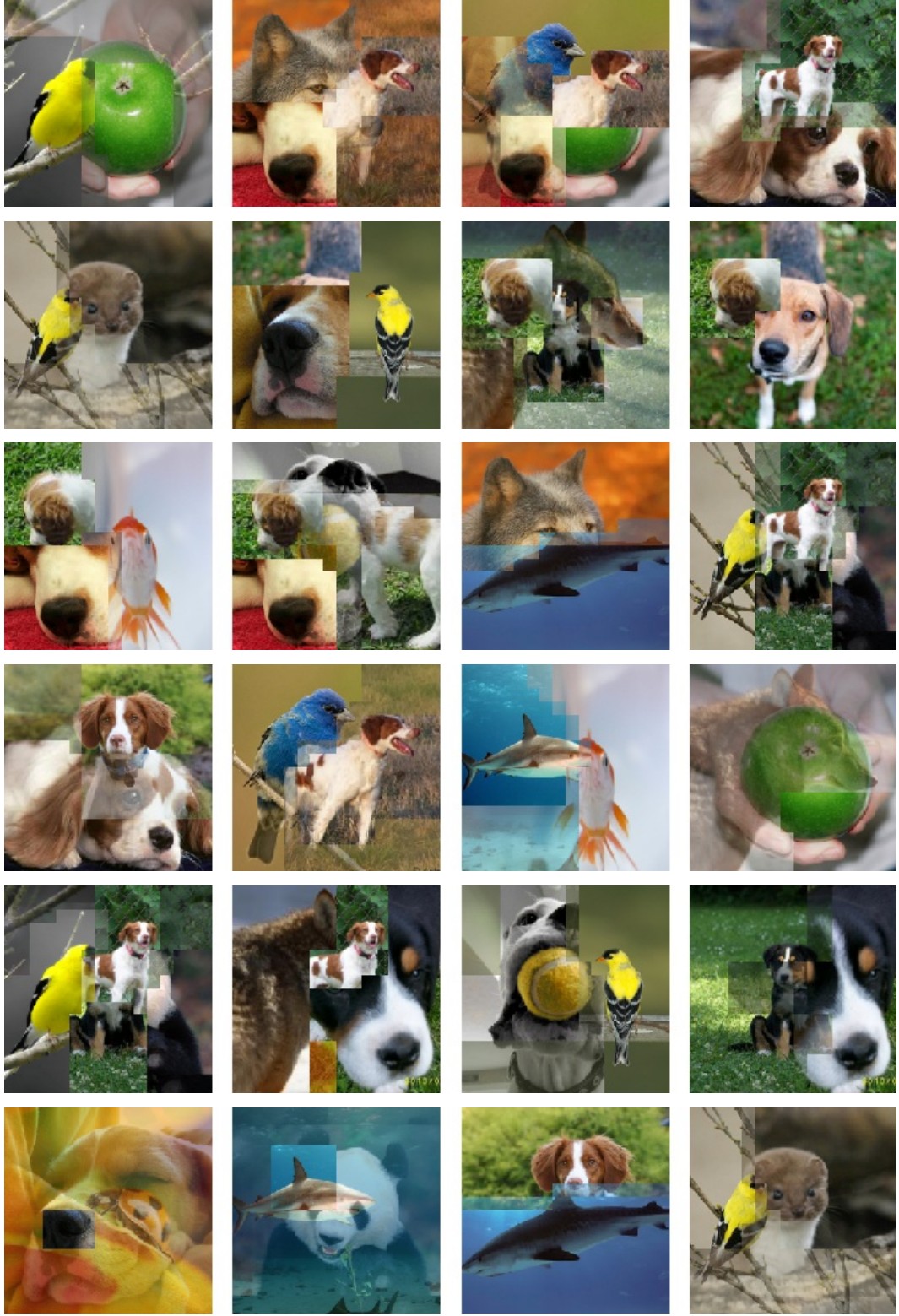

Figure 13: Another mixed output batch with larger $\tau$.

