# OpenReview forum: "Co-Mixup: Saliency Guided Joint Mixup with Supermodular Diversity"
_ICLR.cc/2021/Conference — ICLR 2021 Oral_

### Official Review · AnonReviewer1 · 2020-10-28
**Interesting, but questions about saliency and running time**

**Rating:** 7
**Confidence:** 3

**Review:**

This paper proposes a new mixup method that encourages diversity among the samples mixed from a minibatch of data in addition to saliency of each mixed sample. The authors formulate two objectives: 1. a BP set function (submodular + supermodular), and 2. a submodular relaxation obtained by modularizing the supermodular component. Then they solve this problem approximately with coordinate descent by modularizing wrt the update coordinate at each step. This approach outperforms mixup baselines on image classification and several other tasks (calibration, object localization, and robustness).


The paper presents interesting ideas with impressive accuracy results. My biggest concerns with this work are clarity, thoroughness of experiments, and whether it is too computationally expensive to be used in practice.


Significance:
- The proposed algorithm's running time may be prohibitive for some applications. In the appendix the authors mention partitioning each minibatch and running the algorithm on each partition to make running time feasible, which suggests that accuracy improves at the expense of running time. Section 4.2 presents the algorithm as having linear running time, the exponential dependence on the number of labels $|\mathcal{L}|$ should be mentioned here

Experiments:
- The results sections compare against good baselines across several tasks, but this would be stronger if it compared to non-mixup baselines.
- A more thorough ablation study would analyze each term in the proposed objective function, compare to the mixup baselines applied to m>2 inputs, and compare to the original set modularization method.
- The paper proposes an alternative heuristic to the set modularization method of Narshiman and Bilmes 2005. The claim that Algorithm 1 is faster and produces better solutions would be stronger with a thorough empirical comparison to set modularization.

Clarity:
- Saliency is described at a high level several times, but it's unclear how these values are actually computed in experiments
- Notation for $z$ is somewhat difficult to parse in the statements/proofs of Propositions 1 and 2. $f(z)$ is pairwise supermodular when considering the same column index across 2 different matrices of output coefficients $(z_{j_1,k},z_{j_2,k})$ (Proposition 1), but $f(z)$ is modular when considering the full output label matrix $z_j$ (Proposition 2). What does this mean when the optimization is over all matrices $z$? Do the ground set and constraints change in each case?
- typo: In section 4.2, "proposition 1" should be "Proposition 2"

Organization:
- Sections 5.4 (5.5) on robustness (sensitivity) as written in the main text are not informative. They should either contain more concrete statements about results or be moved entirely to the supplementary material.
- Also, discussion of the ground set would be much clearer if it appeared before the bottom of page 6

Pros
- Improves over previous strong mixup baselines
- The proposed method outperforms baselines even when m=2 (it's worth mentioning this in the main text even if there is not enough room for a full ablation)

Cons
- Running time of the algorithm may be prohibitive in certain applications, this should be analyzed an discussed
- Several issues with clarity and organization (see above)


Questions:
- Does larger m always improve accuracy, or does this eventually decrease?
- Exactly how is saliency computed in the experiments? Is it the same pretrained map or does it vary with architecture?
- How much time does co-mixup take compared to training the network?
- Are Algorithm 1 and applying mixup ideas to m>2 samples novel contribution?


EDIT: The author response addressed all my concerns and answered all my questions, in particular that the exponential running time is a worst-case bound that is indeed loose in practice. I believe that the revised version will be much clearer and am therefore increasing my score

---

> ### Author Response · Authors · 2020-11-19
> **Response to AnonReviewer1 (1)**
>
> ##### **Algorithm's running time and time complexity**
>
> Answer) Thank you for your comment. Based on our implementation, we train models with Co-Mixup in a feasible time. For example, in the case of ImageNet training with 16 Intel I9-9980XE CPU cores and 4 NVIDIA RTX 2080Ti GPUs, Co-Mixup training requires 0.964 s per batch, whereas the vanilla training without mixup requires 0.374 s per batch (about 2.5 times faster). Note that Co-Mixup requires saliency computation, and when we compare the algorithm with Puzzle Mix, which performs the exact same saliency computation, Co-Mixup is only slower about 1.04 times (i.e., almost same processing time). Besides, as Co-Mixup down-samples the data to the fixed size regardless of the data dimension, the additional computation cost of Co-Mixup relatively decreases as the data dimension increases. Moreover, the additional computation time of Co-Mixup can be easily adjusted by controlling parameters (such as m). We wish to emphasize that the training procedure needs to be done once and our inference procedure is a simple forward pass same as vanilla classifiers. Having said that, we believe improving the training time and speed is an important future work.
>
> We provide more detailed information about the processing time of Co-Mixup in the supplementary material Section 3- Figure 2 of the revision, e.g., empirical time complexity over the number of inputs m and the number of labels $|\mathcal{L}|$. Note that what we mentioned (e.g., the exponential dependence on the number of labels $|\mathcal{L}|$) are the theoretic worst-case complexity of alpha-beta swap algorithm [1], and empirically the alpha-beta swap algorithm is much faster than the theoretic worst-case complexity (see Figure 15 of [1] which shows almost linear time complexity). When we test our implementation of Algorithm 1 over 100 trials, it has almost linear time-complexity on $|\mathcal{L}|$ (see supplementary material Section 3- Figure 2 left). We clarify the time complexity analysis in the supplementary material Section 3 at the revision by clearly stating terms including ‘worst-case’.
>
> [1] Y. Boykov, O. Veksler and R. Zabih, "Fast approximate energy minimization via graph cuts," in IEEE Transactions on Pattern Analysis and Machine Intelligence, vol. 23, no. 11, pp. 1222-1239, Nov. 2001
>
>
> ##### **Comparison with non-mixup baselines**
>
> Answer) Thank you for the suggestion. We perform experiments with VAT regularization [2] on CIFAR-100 with PreActResNet18 for 300 epochs in the supervised setting. We tune $\alpha$ (coefficient of VAT regularization term) in {0.001, 0.01, 0.1}, $\epsilon$ (radius of $\ell$-inf ball) in {1, 2}, and the number of noise update steps in {0, 1}. As a result, we find the best Top-1 error rate of 23.14% that Co-Mixup (19.87%) significantly outperforms. We include the result in the supplementary material Section 6.4.
>
> [2] Miyato et al., Virtual adversarial training: a regularization method for supervised and semi-supervised learning, IEEE transactions on pattern analysis and machine intelligence, 41(8):1979–1993, 2018.
>
> ##### **Ablation study about mixup baselines applied to m>2 inputs**
> Answer) Thank you for the suggestion. To further investigate the effect of the number of inputs for mixup in isolation, we conduct an ablation study on baselines using multiple mixing inputs (CIFAR-100,  PreActResNet18). For fair comparison, we use Dirichlet($\alpha$, ..., $\alpha$) prior to sample the mixing ratio, and we tune $\alpha$ in {0.2, 1.0, 2.0}. As a result (see the table below), we find that mixing multiple inputs decreases performance gains of each mixup baseline and is still inferior to the performance of Co-Mixup (19.87%). These results demonstrate that mixing multiple inputs could lead to possible degradation and support the necessity of considering saliency information and diversity as in Co-Mixup. We include this result in Section 5.5 of the revision.
>
> | # inputs for mixup     |        Input       |       Manifold      |      CutMix       |
> |--------------------------|:--------------:|:-----------------:|:---------------------:|
> |# inputs = 2           |  22.43 %  | 21.64  %   | 21.29 %|
> |# inputs = 3           |  23.03 %  | 22.13  %   | 22.01 %|
> |# inputs = 4           |  23.12 %  | 22.07 %    | 22.20 %|
>
> <Table. # inputs vs Top-1 error rate for mixup baselines. The lower the better.>

---

> ### Author Response · Authors · 2020-11-19
> **Response to AnonReviewer1 (2)**
>
> ##### **Empirical comparison to set modularization (Narshiman and Bilmes 2005)**
>
> Answer) Thank you for the suggestion. We compare the proposed algorithm and the BP algorithm proposed by Narshiman and Bilmes 2005 by performing the following experiment with 100 random seeds. In detail, we evaluate function values of solutions by each method with a unary cost matrix sampled from the uniform distribution. We compare methods over various scales by controlling the number of mixing inputs m.
>
> The table below shows the averaged function values with standard deviations in the parenthesis. As we can see from the table, the proposed algorithm achieves much lower function values and deviations than the method by Narshiman and Bilmes over various scales. Note that the method by Narshiman and Bilmes has high variance due to randomization in the algorithm.
>
> |       Algorithm          |    m=5    |     m=10       |       m=20       |       m=50         |          m=100      |
> |:--------------------------|:--------------|:-----------------|:---------------------|:-----------------|:---------------------|
> |        Ours                   |  3.1 (1.7)  |      15 (6.6)   |       54 (15)    |         205 (26)    |           469 (31)     |
> |        Narasimhan      |  269 (58)  |  1071 (174)  |    4344 (701) |    24955 (4439) |       85782 (14337)   |
> |        Random guess  |  809 (22)  |   7269 (92)   |  60964 (413) |  980973 (2462) |    7925650 (10381)   |
>
> <Table. Performance comparison with the set modularization method by Narshiman and Bilmes. The lower the better>
>
> We further compare the algorithms’ convergence time (s) in the table below. The experiments verify that the proposed algorithm is much faster and effective than the method by Narshiman and Bilmes. We include this result in the supplementary material Section 4.2 of the revision.
>
> |        Algorithm          |     m=5     |     m=10       |       m=20       |       m=50         |           m=100       |
> |:--------------------------|--------------:|-----------------:|---------------------:|-----------------:|---------------------:|
> |    Ours             |  0.02 |  0.04  |   0.11  | 0.54  |    2.71 |
> |   Narasimhan |  0.06 |  0.09  |   0.27   | 1.27  |    7.08 |
>
> <Table. Convergence time (s) comparison with the set modularization method by Narshiman and Bilmes.>
>
>
> ##### **Clarity and a question about saliency computation**
>
> Answer) We calculate the gradient values of training loss with respect to the input data and measure the $\ell_2$-norm of the gradient values across input channels following the practice in [1] and Puzzle Mix (described in the supplementary Section 3). This is one of the simplest methods and does not require any additional architecture dependent modules for saliency calculation. For clarity, we describe this in Section 4.1 of the revision.
>
> [1] Simonyan et al., deep inside convolutional networks: Visualising image classification models and saliency maps, arxiv, 2013
>
> ##### **A question about Proposition 1 and 2 (modularity and supermodularity)**
>
> Answer) Note that Proposition 2 is about the compatibility term “without the clipping” (i.e., max function in Equation 1), whereas Proposition 1 is about the compatibility term “with the clipping”. The supermodularity depends on the existence of clipping. We make this statement more clear in the revision Section 4.2.
>
> ##### **Comments for organization**
> Answer) Thank you for the pointer. According to your suggestion, we move the robustness table in the main text and move the sensitivity subsection to the supplementary material Section 6.3. In  addition, we include a more detailed description of the ground set in Section 4.2- Analysis paragraph.
>
> ##### **“Does larger m always improve accuracy, or does this eventually decrease?”**
>
> Answer) Thank you for your question. We train Co-Mixup models with larger m and find that the performance is saturated. Specifically, on CIFAR-100 with PreActResNet18, Co-Mixup with m=50 achieves 19.96 Top-1 error rate  (averaged over three random seeds) whereas m=20 achieves 19.87. We add this result in our revision supplementary material Section 6.3.
>
> ##### **“Are Algorithm 1 and applying mixup ideas to m>2 samples novel contributions?”**
>
> Answer) To the best of our knowledge, existing mixup based methods only tackle how to mix a given random pair of inputs, and Co-Mixup is the first to attempt considering how to optimally mix various fragments of data encouraging supermodular diversity and craft meaningful supervisory mixup data through a principled optimization formulation. We believe this question is a critical issue in the subfield of mixup. In addition, Algorithm 1 is effective and applicable to a large family of functions and diverse tasks, which has not been introduced before in the data augmentation area.

---

### Official Review · AnonReviewer2 · 2020-10-28
**Review for Co-MIXUP**

**Rating:** 7
**Confidence:** 4

**Review:**

1) Summary
- Mixup is one of the representative data augmentation techniques to improve the generalization of the network.
- The authors proposed the co-mixup technique which is novel.
- Especially, the technique can mix several images with z, and the z is found by optimizing their objective function.
- It is novel, and the experimental results are convincing.

2) Strong points
- co-mixup approach was formulated well
- clearly outperforms the other mixup-like technique such as CutMix and Puzzle Mix

3) Weak points
- Training is slower than others, even if computing z is fast.
- Can we mix more than three images? How about 1000 images?


This paper is a good one, and I look forward to acceptance.

---

> ### Author Response · Authors · 2020-11-19
> **Response to AnonReviewer2**
>
> ##### **“Can we mix more than three images? How about 1000 images?”**
>
> Answer) Thank you for the question. As shown in Figure 2-(b), Co-Mixup can use more than three images per mixup output depending on the clipping values $\tau$. For example, we can find a mixup example consisting of four images as shown in supplementary material Figure 8- first row, third column. Co-Mixup finds the optimal matching of images depending on inter-arrangement among a batch of images and automatically determines the number of mixing inputs for each output. According to this, we can mix more than 1000 images using the same algorithm.

---

### Official Review · AnonReviewer4 · 2020-10-29
**interesting paper but requires some clarification**

**Rating:** 7
**Confidence:** 3

**Review:**

This paper proposes a new batch mixup method, co-mixup, to improve the networks’ generalization performance and robustness. It formulates the construction of a batch of mixup data by maximizing the data saliency measure of each individual mixup data and the supermodular diversity among the constructed mixup data. An iterative submodular minimization algorithm is used to solve the proposed problem through approximation. Promising empirical performance is reported on several tasks.

All previous mixup methods are limited to mixup example generation between a pair of input examples. This paper can be viewed as an extension of the Puzzle mixup, and it generates a mixup example over multiple input examples. The optimization problem formulated seems to be very reasonable by maximizing the saliency and diversity of the mixup data. The proposed method also demonstrates slightly better performance than other mixup methods.  Overall, this is an interesting paper.

There are however several issues that are not clear. The authors can clarify the following questions:

1.	How are the labels of the mixup examples determined?  Will each generated example become a multi-label example? How to perform training with the generated data?

2.	As the proposed method requires additional process to produce information such as the saliency information and compatibility information (the matrix A), won’t this induce additional computational cost?
3.	As stated in the paper, the A_c matrix measures the distances between locations of salient objects in the input examples. Does this require object localization? Or it simply computes the distances between the feature locations?
4.	How much does the \omega value (to compute A) affect the performance of the proposed approach?

5.	In Section 4.2, two criteria are described and then an approximation is proposed based on the two criteria.  Although there is nothing wrong about the criteria, it is unclear how good can an approximation be by simply satisfying these two criteria? That is, what is the quality of the approximation to the original problem?  Can the approximation guarantee an optimal solution to the original problem?

6.	In the experiments, although the proposed approach demonstrates promising results, the differences between the proposed method and the comparison method, Puzzle Mix, are very small. The paper claims Co-Mixup significantly outperforms all other baselines. How significant are the differences in Table 1?

7.	The paper limits the comparison to mixup methods. How is the performance level of the co-mixup by comparing with other regularization based methods? For example, VAT regularizaiton based methods.

---

> ### Author Response · Authors · 2020-11-19
> **Response to AnonReviewer4**
>
> ##### **“How are the labels of the mixup examples determined?”**
>
> Answer) Thank you for the comment. Given the one-hot target labels $y_B\in$ {0,1}$^{m \times C}$ of the input data with $C$ classes, we generate soft target labels for mixup data as $y_B^\intercal u_j$ for $j=1,\ldots,m'$, where $u_j =\frac{1}{n} \sum_{k=1}^{n} z_{j,k} \in [0,1]^{m} $ represents the input source ratio of the $j^{\text{th}}$ mixup data. We train models to estimate the soft target labels by minimizing the cross-entropy loss. We add this explanation in Section 3 of the revision.
>
> ##### **A question for additional computations of Co-Mixup (saliency and compatibility matrix)**
>
> Answer) We follow the method of [1] and Puzzle Mix to calculate the saliency information. Although this process requires additional computations, the models can be trained within a feasible time with outperforming performance. Note that the computation cost for compatibility information is negligible. We describe this computation in detail in the next question.
>
> [1] Simonyan et al., deep inside convolutional networks: Visualising image classification models and saliency maps, arxiv, 2013
>
> ##### **A question for $A_c$ matrix computation**
>
> Answer) We calculate the $l_1$-distance between the most salient regions for compatibility matrix. In other words, $A_c[i,j] = ||\text{argmax}_k s_i[k] - \text{argmax}_k s_j[k] ||_1$, where $s_i$ is the (smoothed) saliency map of the $i^\text{th}$ input and $k$ is a location index (e.g., $k$ is a 2-D index for image data). We clarify this in Section 4.1- Diversity paragraph of the revision.
>
>
> ##### **How much does the $\omega$ value (to compute A) affect the performance?**
> Answer) Thank you for the suggestion. We perform sensitivity analysis of $\omega \in$ {0, 5e-4, 1e-3, 5e-3, 1e-2} on CIFAR100 with PreActResNet-18. As results, we have comparable performances on {1e-3, 5e-3, 1e-2} (best at 1e-3 which is the original setting), but have inferior performances on {0, 5e-4}, which demonstrates the effectiveness of the compatibility term. We include this result in the revision supplementary material Section 6.3.
>
> ##### **“What is the quality of the approximation/algorithm to the original problem?”**
> Answer) Thank you for the pointer. To inspect the optimality of our algorithm, we perform a comparison experiment (with 100 different random seeds) and compare the function values of the solutions of our algorithm, brute force search algorithm, and random guess. Due to the exponential time complexity of the brute force search, we compare the algorithms on small scale experiment settings. Specifically, we test algorithms on settings of $(m=m'=2, n=4)$, $(m=m'=2, n=9)$, and $(m=m'=3, n=4)$ varying the number of inputs and outputs $(m, m')$ and the dimension of data $n$.
>
> As a result, we find that the proposed algorithm achieves near-optimal solutions over various settings. We also measure relative errors between ours and random guess, and our algorithm achieves relative error less than 0.01. We include this result in supplementary material Section 4.
>
> |    Configuration    |   Ours  | Brute force (optimal) | Random guess     |    Rel. error |
> |:--------------------------|:--------------:|:-----------------:|:---------------------:|:---------------------:|
> |  $(m=m'=2, n=4)$  |   1.91   |            1.90                |            3.54            |     0.004   |
> |    $(m=m'=2, n=9)$  |   1.93   |            1.91                |            3.66            |      0.01    |
>  |   $(m=m'=3, n=4)$  |   2.89   |            2.85                |           22.02           |     0.002   |
>
> <Table: Mean function values of the solutions over 100 different random seeds. Rel. error means the relative error between ours and random guess>
>
> ##### **“How significant are the differences in Table 1?”**
> Answer) Thank you for the question. We run experiments with ten different random seeds to compare the Top-1 error rates between Puzzle Mix and Co-Mixup (CIFAR-100, PreActResNet18, 300 epochs). Based on the results, we perform a paired t-test and observe T-statistics: 2.28 and P-value: 0.02. This result demonstrates that Co-Mixup outperforms Puzzle Mix with statistical significance. Moreover, in Table 1, Co-Mixup outperforms Puzzle Mix over 1.34%p with ResNeXt29-4-24, and we believe this is a significant performance improvement. We include this result in supplementary material Section 6.5.
>
> ##### **Comparison with other regularization based methods**
> Answer) Thank you for the suggestion. We perform experiments with VAT on CIFAR-100 with PreActResNet18 in the supervised setting. We tune $\alpha$ (coefficient of VAT regularization term) in {0.001, 0.01, 0.1}, $\epsilon$ (radius of $\ell$-inf ball) in {1, 2}, and the number of noise update steps in {0, 1}. As a result, we find the best Top-1 error rate of 23.14% that Co-Mixup (19.87%) significantly outperforms. We include the citation and the result in the supplementary material Section 6.4.

---

### Author Response · Authors · 2020-11-19
**Dear reviewers**

Thank you for your time and the effort spent providing feedback. We appreciate the encouraging comments [R1] “The paper presents interesting ideas with impressive accuracy results”. [R2] “The authors proposed the co-mixup technique which is novel”. “approach was formulated well”. [R4] “The optimization problem formulated seems to be very reasonable”, “promising empirical performance is reported on several tasks”, “this is an interesting paper”.

We address your comments and questions in the replies, and revised our submission.

Best,
Co-Mixup authors

---

### Decision · Program_Chairs · 2021-01-07
**Final Decision**

**Decision:**

Accept (Oral)

**Comment:**

This paper proposes a type of Mixup-style data augmentation that works at the batch level rather than simply between pairs of examples. Each generated example accumulates salient images from potentially many other examples while ensuring diversity across the generated examples. This is achieved through a 4-part objective with submodular and supermodular components. The paper demonstrates the method using extensive experiments, including generalization performance on CIFAR-100, Tiny ImageNet, ImageNet and GoogleCommands. It also explores weakly supervised object localization, expected calibration error, and robustness to random replacement and Gaussian noise.

Reviewer 1 thought the approach was interesting but raised some concerns with clarity, thoroughness of experiments and whether the approach was computationally prohibitive to be used in practice. I was surprised myself that a discussion on the trade-off between computational expense and accuracy gain was not discussed in the submission. The authors responded to the review, adding a comparison to the BP algorithm (Narshiman and Bilmes 2005). The empirical result seems to back up the claim that the proposed algorithm finds a better solution and with less variance. It also appears to run much faster. The authors also responded to minor issues raised with respect to clarity and organization. In their response, the authors provided considerable detail with respect to running time and time complexity, and show that models trained with co-mixup are practical, though they do come with a significant added cost. The authors added the requested comparisons to non-mixup baselines and enhanced the ablation study. In my opinion, this is a comprehensive and satisfying response, and the paper has improved in many respects since submission.

The review from R2 was largely positive, though limited in its scope. They also expressed concerns with training time (addressed in the response to R1). Clearly the approach extends to an arbitrary (m) number of images; this was explicit in the paper/formulation and clarified by the authors. I have some concern that R2 may have skimmed the paper if they missed this point.

Reviewer 4 thought the paper was interesting and asked several clarifying questions. They expressed concern with the significance of the reported gains. Similar to R1, they asked about non-mixup baselines (VAT specifically). This was addressed in the response to R1. The authors responded to the clarifying questions and addressed the issue of significance.

Like the reviewers, I think that this is an intriguing, fresh, and elegant way to perform data augmentation. I appreciate that it has been evaluated just not from the pure generalization setting, but from other angles like robustness and calibration. There are still some outstanding concerns regarding the computational effort required to use Co-Mixup, so this would be nice to see in follow-up work.